

# Glacial isostatic adjustment strain rate - stress paradox in the Western Alps, impact on active faults and seismicity

Juliette Grosset[1], Stéphane Mazzotti[1], Philippe Vernant[1]

[1]Géosciences Montpellier, Université de Montpellier, CNRS, Montpellier, 34000, France

*Correspondence to*: Juliette Grosset (juliette.grosset@umontpellier.fr)

**Abstract.** In regions formerly glaciated during the Last Glacial Maximum (LGM), Glacial Isostatic Adjustment (GIA) explains most of the measured uplift and deformation rates. GIA is also proposed as a key process contributing to fault activity and seismicity shortly after the LGM and potentially up to present-day. Here, we study the impact of GIA on present-day fault activity and seismicity in the Western Alps. We show that, in the upper crust, GIA induces horizontal compressive stress

perturbations associated with horizontal extension rates. The latter agree with the observed geodetic strain rates and with the seismicity deformation patterns. Yet, in nearly all cases, the GIA stress perturbations tend either to inhibit fault slip, or to promote fault slip with the wrong mechanism compared to the seismicity deformation style. Thus, although GIA from the LGM explains a major part of the geodetic strain rates, it does not drive nor promote the observed seismicity (which must be driven by other processes). This apparent strain rate - stress paradox results from the gradual diminution over time of the finite

shortening induced in the upper crust by the LGM icecap. A direct corollary of our results is that seismicity and seismic hazard studies in the Western Alps cannot directly integrate geodetic velocities and strain rates, but instead require detailed modeling of the GIA transient impact.

## 1. Introduction - GIA, deformation and seismicity

Glacial Isostatic Adjustment (GIA) is the mechanical response of the Earth's crust and mantle to loading / unloading

cycles of continental ice sheets and glaciers. The associated surface deformation is observed in geomorphological features such as raised paleo-shorelines and in geodetic measurements of present-day uplift rates or horizontal strain rates (cf. review in Peltier et al., 2022). The impact of GIA and the associated stress perturbations on fault activity and seismicity has been studied since the seminal work of Johnston (1987), Quinlan (1984) and Stephansson (1988), up to recent developments including more complex Earth and fault mechanics models (cf. Steffen et al., 2021). A common feature of these studies is the

demonstration that GIA increases fault activity and seismicity shortly after the main deglaciation phase in regions formerly glaciated during the Last Glacial Maximum (LGM) (eg., Hetzel and Hampel, 2005; Muir-Wood, 2000; Steffen et al., 2014; Stewart et al., 2000; Wu et al., 1999). In contrast, the potential effects of GIA stress perturbations on present-day faulting and seismicity in and near formerly glaciated regions remains debated (e.g., Bungum et al., 2010; Bungum and Eldholm, 2022; Brandes et al., 2015; Grollimund and Zoback, 2001).



30   The first-order mechanics of GIA-related seismicity involves lithosphere flexure coupled with mantle relaxation in response to ice loading and unloading. Associated stress perturbations in the upper crust can reach a few MPa, sufficient to induce rupture on faults near failure equilibrium or to unclamp faults thus allowing the release of long-term stored tectonic stress (e.g., Craig et al., 2016; Hetzel and Hampel, 2005; Steffen et al., 2014). These effects are based on the same model (Fig. 1), wherein the ice loading results in a downward flexure and a horizontal compressive stress perturbation in the upper half of

35 the elastic lithosphere beneath the load (resp. upward flexure / extensive stress in the forebulge regions). Following deglaciation, the lithosphere unbending is dampened by the mantle viscosity resulting in a gradual diminution of the initial bending stress. The associated strain corresponds to a maximum shortening at the peak of glaciation followed by a diminution of shortening during the postglacial phase. As a result, the present-day surface strain rate corresponds to an extension rate while the stress perturbation remains compressive. This apparent contradiction is similar to the strain rate - stress paradox

40 observed in subduction zone forearcs in relation with transient interseismic locking of the megathrust fault (Wang, 2000). At the end of the GIA cycle, bending stress are fully released and the plate regains its initial background state of stress (plus the potential steady-state tectonic loading) (Fig. 1).

   Owing to GNSS (Global Navigation Satellite Network) data, present-day horizontal extension rates due to GIA have been measured with increasing accuracies not only in regions formerly covered by LGM ice sheets (Calais et al., 2006; Keiding

45 et al., 2015; Tarayoun et al., 2018), but also in regions of smaller LGM icecaps and mountains glaciers such as the European Alps (Masson et al., 2019; Nguyen et al., 2016; Walpersdorf et al., 2018). These horizontal strain rates have been compared with seismicity deformation patterns and rates, with various degrees of agreement in both styles and amplitudes (Keiding et al., 2015; Mazzotti et al., 2005; Sánchez et al., 2018). One of the goals of these comparisons is the integration of GNSS velocities and strain rates in seismic hazard models (Mathey et al., 2020), assuming that these short-term (5–20 yr) data can

50 provide information on longer-term ($10^2$–$10^5$ yr) earthquake activity. However, the apparent GIA strain rate - stress paradox puts strong doubts on the pertinence of using GNSS rates to compare with seismicity in regions affected by ongoing transient GIA deformation (even more so for regions with very low tectonic loading rates such as intraplate domains).

   In this study, we compare GIA deformation and stress predictions with present-day GNSS strain rates, fault activity and seismicity in the Western Alps (France, Italy, Switzerland), where GIA contributes to a large part of the measured

55 geodetic deformation rates (Sternai et al., 2019; Stocchi et al., 2005). Specifically, we analyze the effect of GIA stress perturbations on a series of typical fault systems of the Western Alps, in order (1) to compare with their observed kinematics and associated earthquake focal mechanisms, and (2) to test whether GIA stresses tend to promote or inhibit the present-day seismicity.

60



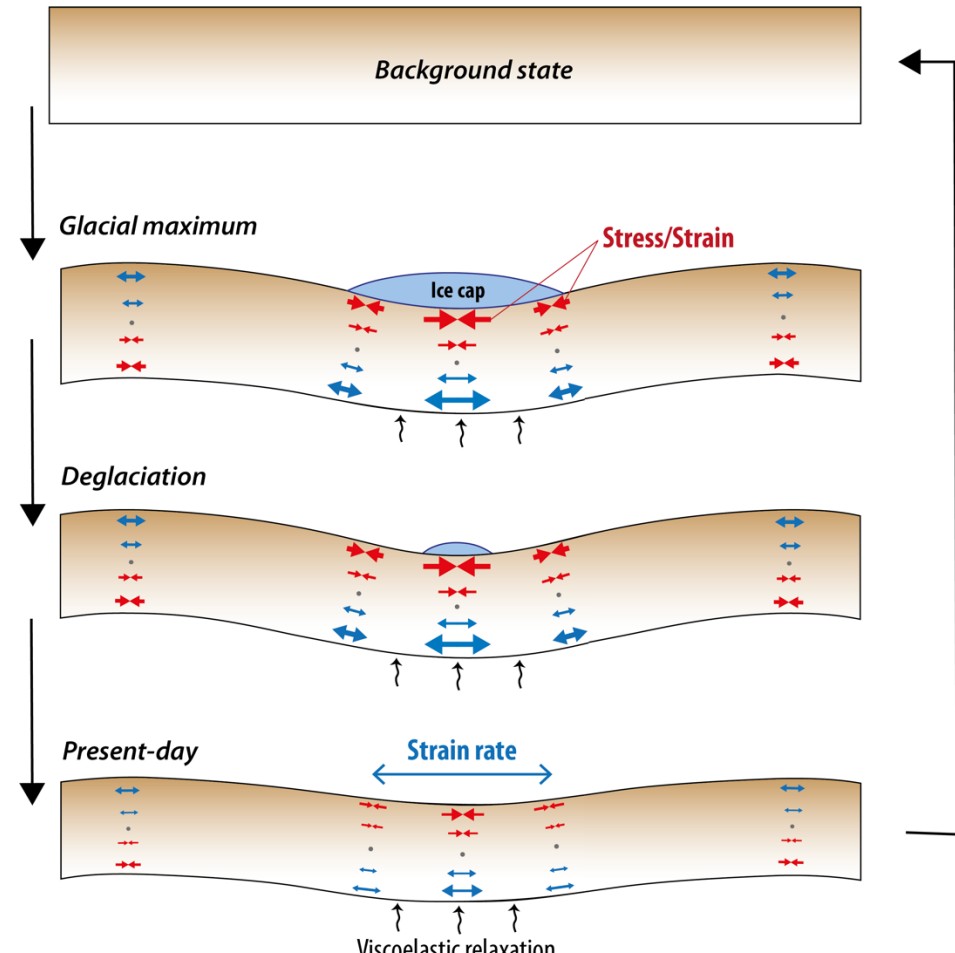

**Figure 1: Conceptual model of Glacial Isostatic Adjustment (GIA) induced stresses, strains and strain rates**

## 2. Western Alps seismicity and GNSS strain rates

The Western Alps (Fig. 2) are one of the most seismically active regions in western Europe (Larroque et al., 2021).
Background seismicity affects the whole region, with a high density of small to medium earthquakes (Mw ≥ 2.5) in the high-altitude inner parts of the mountain range (Fig. 2b). In addition, several larger damaging earthquakes have occurred in the Western Alps over the last centuries (e.g., Epagny-Annecy, 1996, Mw = 4.9; Ligurian-Imperia, 1887, Mw = 6.8; Bale, 1356, Mw = 6.5). In the high-altitude inner areas, strain patterns derived from earthquake focal mechanisms show an overall normal-faulting style, with an extension direction perpendicular to the Alpine arc (Fig. 2b). These normal-faulting earthquakes can be divided into two N-S clusters: a western cluster, roughly in the Briançonnais region, with earthquake depths concentrated ca. 5–10 km, and a smaller eastern cluster, with deeper focal depths ca. 10–20 km (Mathey et al., 2021). The surrounding lower



altitude and foreland regions show a mix of strike-slip and reverse faulting with a general arc-perpendicular compression (Fig. 2b; Delacou et al., 2004; Mathey et al., 2021; Mazzotti et al., 2021; Sue et al., 1999). In general, only a few known fault structures can be associated with observed seismicity, e.g. the Belledonne Fault (Thouvenot et al., 2003a), the Moyenne Durance Fault (Cushing et al., 2007), or the Vuache Fault (Baize et al., 2011). In contrast, most of the normal-faulting earthquakes in the inner massifs (above the Penninic Frontal Thrust) are not directly related to know major faults or structures (Larroque et al., 2021; Sue and Tricart, 2003).

**Figure 2: Western Alps tectonic and seismicity setting.** (a) Instrumental seismicity of western Europe (catalogs: SHARE, Stucchi et al., 2013). Predicted Adria micro-plate motion relative to Eurasia represented along the Adria border. (b) Instrumental seismicity in Western Alps (Manchuel et al., 2018) and interpreted seismic strain styles from focal mechanisms (Rabin et al., (2018), Thouvenot et al., (1998,2003) , Cushing et al. (2008), Sue et al., (2003, 1999), Matthey et al. (2021)).

The present-day geodynamics of the Central and Eastern Alps are primarily controlled by the counter-clockwise rotation of the Adria microplate relative to Eurasia, with a rotation pole east of the Western Alps near Turin, northwestern Italy (Fig. 2a; Battaglia et al., 2004; D'Agostino et al., 2008). However, this rotation kinematics is incompatible with the seismicity



and geodetic horizontal deformation patterns in the Western Alps (Fig. 2.b). GNSS data indicate that the overall horizontal deformation across the Western Alps is smaller than their current resolution, i.e., less than 0.3 mm.yr$^{-1}$ relative motion between northwestern Italy (Pô Plain) and the Rhône Valley - Eastern France (Masson et al., 2019; Sánchez et al., 2018).

Geodetic data also indicate that the Western Alps are affected by a significant regional uplift rate between 0.5 and 2.5 mm.yr$^{-1}$ peaking in the inner high-altitude areas and decreasing towards 0 mm.yr$^{-1}$ in the surrounding lowlands (Fig. 3a; Brockmann et al., 2012; Masson et al., 2019; Nguyen et al., 2016; Nocquet et al., 2016). The highest uplift rates roughly coincide with horizontal arc-perpendicular extension rates ca. 2 x 10$^{-9}$ yr$^{-1}$ in the inner regions (Switzerland, French-Italian border, Fig. 3b), while the surrounding lowlands show horizontal arc-perpendicular shortening rates ca. (1–2) x 10$^{-9}$ yr$^{-1}$ (Jura, Rhone Valley, Fig. 3b) (Masson et al., 2019; Nguyen et al., 2016; Sánchez et al., 2018; Walpersdorf et al., 2018). The southern Western Alps are associated with a mix of extension and strike-slip rates ca. (1–2) x 10$^{-9}$ yr$^{-1}$.

These strain rates amplitudes are at the limit of resolution of GNSS techniques and can only be identified through spatial low-pass filtering to remove short-wavelength noise. In a detailed analysis of real and synthetic data using a spatial Gaussian filter, (Masson et al., 2019) show that a filter half-width of 70–100 km provides the best combination of spatial resolution and noise reduction for data in France and the Western Alps. This filter allows the identification of horizontal strain rate signals with a spatial coherence of ca. 100–200 km and a formal 95% confidence interval ca. (0.2–1.0) x 10$^{-9}$ yr$^{-1}$, depending on the area, data quality, and network geometry. Considering the dimensions of the Western Alps, their icecap during the LGM, and the lithosphere elastic thickness (cf. Section 3), 100–200 km is a reasonable estimation of the expected wavelength of the GIA signal. Thus, in the flowing, we apply the same 90-km-half-width Gaussian filter on GNSS velocities and GIA model velocities for strain rate comparisons. Details of the method and additional strain rate maps at different filtering half-widths can be found in the Supplementary Material 1 for reference.

This overall deformation pattern (slow horizontal extension rates coupled with relatively fast uplift rates) is at the core of the current debate on the processes responsible for the ongoing geodynamics of the Western Alps. Because the role of regional plate tectonics is very small or null, recent studies consider alternative processes including mantle and slab-tear dynamics (Sternai et al., 2019), surface erosion (Champagnac et al., 2007; Vernant et al., 2013), or GIA (Chéry et al., 2016; Mey et al., 2016). Numerical modeling shows that the isostatic response to erosion can generate the observed extension strain rate pattern, but the associated uplift rates are significantly smaller than the GNSS velocities (Sternai et al., 2019; Vernant et al., 2013). On the contrary, GIA from the Last Glacial Maximum can explain the present-day uplift rates (Chéry et al., 2016; Mey et al., 2016), but a detailed comparison with horizontal deformation is lacking. In the following section, we test the compatibility of GIA models with present-day vertical and horizontal deformation rates of the Western Alps.





**Figure 3. Western Alps geodetic and GIA deformation rates.** (a) Vertical velocity field from GNSS permanent networks (Masson et al., 2019). (b) Horizontal strain rate field extracted from Gaussian filter method (cf. text). (c) Best-fit GIA model strain rate field ($h_e$ = 20 km, $\tau$ = 5500 yr). Strain rates are color-coded according to the deformation style. Grey line is Last Glacial Maximum ice extension (Mey et al., 2016).

## 3. GIA models

### 3.1 Model setup

Due to the time scale of glacial loading / unloading cycles ($10^2$–$10^5$ yr), GIA mechanical models consider both elastic and viscous deformation of the Earth's crust and mantle, with a large array of Earth rheology hypotheses and simplifications. In its simplest form, the surface response to GIA can be modeled as that of a thin elastic plate overlying a linear viscous fluid (e.g., Mey et al., 2016). The more complex and most common modeling approach involves a 1D rheology stratification assuming an upper elastic layer, commonly 60–100 km thick, overlying multiple mantle layers with different Maxwell visco-elastic properties (cf. Peltier and Andrews, 1976; Spada et al., 2011). Higher levels of complexities can include asymmetrical 3D variations of rheology (elastic thickness and viscous properties, (e.g.,Bagge et al., 2021; Steffen et al., 2006; Wu, 2005),



inclusion of viscous bodies within the elastic layer (e.g., Klemann and Wolf, 1999; Wu and Mazzotti, 2007), and non-linear or transient viscous rheologies (e.g., Giunchi and Spada, 2000; Lau et al., 2021; van der Wal et al., 2013).

      Owing to the relative simplicity of both its parameterization and computation requirements, the 1D, symmetrical, Maxwell, layered model (hereafter "1D Maxwell") is the standard approach in most GIA studies. However, studies testing more complex rheology structures have shown that their predictions can deviate substantially from those of the "1D Maxwell"

model, with better fit to the surface observables in several cases (cf. Bagge et al., 2021; Lau et al., 2021; Steffen et al., 2006; van der Wal et al., 2013). Of particular note is the integration of non-linear transient mantle rheologies, which are also proposed to explain post-seismic relaxation following major earthquakes (e.g., Freed et al., 2010; Qiu et al., 2018). The integration of such time-dependent viscous rheologies in GIA models would imply significant biases in the standard "1D Maxwell" viscosity estimations (Lau et al., 2021), but would also challenge the precept of stress advection / migration (Steffen et al., 2015; Wu,

1992), which directly results from the Maxwell body approximation for the Earth's mantle.

      In this study, we consider the simplest form of GIA model predictions and use a thin elastic plate overlying a viscous substratum. The elastic plate flexure ($w$) in response to the ice load is computed using the Kirchhoff-Love thin-plate analytic solution (Turcotte and Schubert, 2002; Wickert, 2016), with a flexural rigidity defined by the plate thickness ($h_e$) and its elastic parameters (Young modulus $E = 10^{11}\ Pa$, Poisson coefficient $v = 0.25$). The elastic plate is laterally infinite. The flexure is

computed from a superposition of analytical solutions (using gFlex code, Wickert, 2016). Bending strains and stresses in the elastic plate are computed from the second spatial derivative of the flexural response based on the Kirchhoff-Love theory (no vertical shear stress, only in-plane compression / tension stress). The time dependence of the response due to the upper mantle viscous relaxation is computed assuming an exponential decay controlled by a relaxation time ($\tau$):

$$w(x, y, t) = w(x, y)\ e^{-t/\tau} \tag{1}$$

For the ice model, we use the icecap reconstruction at LGM from (Mey et al., 2016) based on an ice-flow model constrained by geomorphological ice extent and thickness data. The ice model covers the whole Alps, with a mean (resp. maximum) thickness at LGM ca. 21 kyr BP of 415 m (resp. 2445 m). In the Western Alps, the main ice load is located in the major valleys, up to 1000-1500 m ice thickness. Additional smaller ice lobes (few 100 m) were located in the Jura. Keeping in line with the simple modeling approach, we assume (1) that the Alpine icecap reached isostatic equilibrium at LGM, owing

to the relatively short relaxation time of the viscous mantle (cf. Section 3.2) and (2) an instantaneous deglaciation of the icecap at 15 kyr BP, considering that 80% has melted between 21 and 18 kyr BP and the remaining has disappeared by 12 kyr (Ivy-Ochs et al., 2008).

      Using this simplified approach is justified by several points: (1) We are only interested in the present-day state of stress and deformation, to compare with seismicity and geodetic data. Time variations of GIA responses since the LGM are

not relevant to our study, as long as they reach similar present-day states. (2) Alpine GIA models can only be constrained using present-day velocities and strain rates. Thus, the path leading to this present-day state (and thus the dependency on rheology assumptions) are unconstrained. (3) The Western Alps icecap during the LGM has characteristic dimensions of a few 100s km



(Fig. 3). This relatively small load dimension only excites upper mantle relaxation and is thus insensitive to deep mantle rheologies (Steffen et al., 2015). (4) Small (few kyr) time variations in the deglaciation history only impact our results at
present-day by a few percent owing to the mantle relaxation time (cf. Section 3.2). These justifications for our simple approach are illustrated in the Supplemental Material 2, in which we compare its results with those a standard "1D Maxwell" model.

### 3.2 GIA velocities and strain rates

The analytical model allows testing the sensitivity of predicted present-day GIA strain rates and stresses to the large range of assumed elastic plate rigidity and mantle relaxation time. Uplift rate predictions for models with a plate thickness (5
$\leq h_e \leq 100$ km) and a relaxation time ($2000 \leq \tau \leq 20000$ yr) are compared with GNSS uplift rates, assuming that the latter are largely due to GIA (Mey et al., 2016; Nocquet et al., 2016; Sternai et al., 2019). Using F-test statistics on these comparisons, we estimate a best-fit set of ($h_e$, $\tau$) values to $h_e = 10$–20 km and $\tau = 4500$–5500 yr. However, because present-day uplift rates may be impacted by other processes, in the following analysis we allow for a slightly larger range of values: $5 \leq h_e \leq 40$ km and $3000 \leq \tau \leq 7000$ yr (in the following, computations and figures are presented for 72 GIA models with ($h_e$, $\tau$) values
distributed evenly within these ranges). These parameter values indicate a fast and short-wavelength response to the Last Glacial Maximum, in agreement with the results of more complex models (Chéry et al., 2016; Mey et al., 2016).

All retained models predict similar surface horizontal strain rate patterns, with N-S to NNW-SSE extension rates in the inner Western Alps (Fig. 3 and Supp. Mat. 5). These present-day extension rates are maximum in southwestern Switzerland and in the northeastern French Alps, reaching ca. (1–2.5) x $10^{-9}$ yr$^{-1}$. The forebulge regions (Rhone Valley, southern French
Alps, eastern France) are associated with smaller shortening rates ca. (0.2–0.6) x $10^{-9}$ yr$^{-1}$, with a shortening direction perpendicular to the Alpine arc. Transitional areas in between these extension and shortening domains show a variety of extension, strike-slip and shortening rates whose orientations and magnitudes vary strongly with the distance relative to the LGM icecap and with the model parameters (Fig. 3 and Supp. Mat. 5). Thus, expected GIA strain rates in this narrow intermediate region are poorly defined.

To a first order, the GIA model strain rates are consistent with the horizontal strain rates derived from GNSS data in the high-latitude inner parts of the Western Alps (Fig. 3b vs. 3c). In the Swiss-French border region, GIA model and GNSS strain rates agree in both orientations (with a small rotation from N-S to NNW-SSE) and in amplitudes within their uncertainties (GIA model rates ca. (0.9–2.1) x $10^{-9}$ yr$^{-1}$ vs. GNSS rates ca. (2.5 ± 0.5) x $10^{-9}$ yr$^{-1}$). Similarly, in the French Alps GIA model rates are (0.4–0.8) x $10^{-9}$ yr$^{-1}$, in agreement with the GNSS rate ca. (0.9 ± 0.4) x $10^{-9}$ yr$^{-1}$. We estimate that GIA accounts for a
minimum of 30% and up to 100% of the GNSS extension rates observed in the inner Western Alps.

In contrast, the GIA models do not reproduce the relatively high GNSS E-W extension and strike-slip rates ca. (1.5 ± 0.5) x $10^{-9}$ yr$^{-1}$ near the southern French-Italian border and in the western Po Plain (Fig. 3b vs. 3c). In this area, located directly south of the Adria / Eurasia rotation pole, the GNSS strain rates may be in part related to Adria micro-plate kinematics that would predict 0.3–0.5 mm.yr$^{-1}$ of E-W extension in southern Western Alps. Similarly, the GIA models do not explain the



GNSS strain rates observed in the French and Swiss Jura and the southern Upper Rhine Graben. There, GIA model strain rates
are consistent in style (shortening rates) and orientations with GNSS data, but their amplitudes are about 3–4 time smaller (Fig.
3b vs. 3c). This difference is too large to be explained by variations in the GIA model parameters (cf. Supp. Mat. 5), suggesting
that the GNSS shortening rates cannot be solely attributed to Alpine GIA and that additional processes must contribute, such
as possibly GIA effects from the Fennoscandian LGM ice sheets (e.g., Nocquet et al., 2005) or the deformation due to a mantle
plume beneath the Eifel volcanic area (Kreemer et al., 2020).

## 4. Present-day GIA stresses, active fault, and seismicity

### 4.1 GIA model vs. fault and seismicity depths

One of the main issues in using GIA models for studying stress perturbations on crustal faults is how the model setup
relates to the actual Earth rheology and in particular its brittle / elastic domains. Although the upper elastic layer in GIA models
is only a mechanical proxy for the crust and lithospheric mantle flexural rigidity, all studies make the simple assumptions that
(1) this elastic layer corresponds to the crust or lithosphere, depending on its thickness, (2) its top corresponds to the Earth
surface, and (3) seismogenic faults are embedded in the elastic layer starting at its top (e.g., Hetzel and Hampel, 2005; Steffen
et al., 2014). More complex models integrating elastic, brittle and ductile behaviors show that, under long-term geological
loads, the lithosphere flexural rigidity resides where its resistance to failure exceeds ca. 10–20 MPa, which, in the upper crust,
corresponds to depths below ca. 1–4 km (Burov and Diament, 1995; Hyndman et al., 2009; Tesauro et al., 2009). To our
knowledge, no similar study exists for shorter-term loads like glaciation cycles. Thus, matching "real Earth" depths, i.e., those
of earthquakes or faults, with GIA model depths remains an issue.

In the Western Alps, most of the seismicity concentrates in the upper 10 km of the crust, except for the deeper cluster
of extension seismicity ca. 15–20 km depth east of the French-Italian border (cf. Section 2; Delacou et al., 2004; Mathey et al.,
2021). In order to match our GIA model stress predictions to fault and seismicity depths, we consider the following points: (1)
The range of GIA elastic plate thickness derived from GNSS data ($h_e$ = 10–20 km, cf. Section 3) corresponds primarily to the
flexural rigidity of the crust. (2) The upper half of the elastic plate corresponds to a "real Earth" depth range starting ca. 1–4
km depth (cf. discussion above) and extending to ca. 6–14 km depth (starting depth plus half the plate thickness). (3) The lower
half of the elastic plate extends from ca. 6–14 km to ca. 11–24 km "real Earth" depths. Thus, to a first order, the shallow (resp.
deep) seismicity level can be matched with the upper (resp. lower) half of the GIA elastic plate. In the following, we use the
maximum GIA stress perturbation derived at the top (resp. bottom) of the elastic plate to discuss the impact on fault rupture,
keeping on mind that the closer to the plate center, the smaller the GIA induced stress perturbation.

### 4.2 Present-day GIA Coulomb Failure Stress perturbations

As expected from the conceptual flexural model (Fig. 1), shallow GIA stress perturbations correspond to horizontal
compressive stresses below the former Alpine icecap, including at present-day (Fig. 4a) where uniaxial NNW-SSE



compression of 2–3 MPa dominates in southern Switzerland up to the French and Italian borders, rotating to E-W compression of 1–2 MPa in the French Alps. These present-day shallow compressive horizontal stresses coincide directly with the region of current extension rates observed in GNSS (cf. Fig. 3b vs. 4a).

**Figure 4. Present-day GIA stress perturbations and predicted fault-slip rakes in the northern Western Alps.** (a) Horizontal GIA stress perturbations (model $h_e$ = 20 km, $\tau$ = 5500 yr) and faults tested in the Coulomb Failure Stress perturbation analyses. (b) Fault-slip rakes predicted by GIA stress perturbations (boxplot) and observed from earthquake focal mechanisms (black cross; Billant et al., 2015; Delacou et al., 2004; Rabin et al., 2018; Sue et al., 1999; Thouvenot et al., 2003). Boxplots represent predictions for 72 GIA models distributed in (5 ≤ $h_e$ ≤ 40 km) and (3000 ≤ $\tau$ ≤ 7000 yr), combined with three possible fault dip (IMNF: 25°, 45°, 65°; Belledonne: 65°, 70°, 85°; Vuache: 60°, 80°, 80°).

The impact of these stress perturbations on a given fault can be estimated using the variation of Coulomb Failure Stress ($\Delta_{CFS}$) derived from the fault geometry and the associated fault shear and normal stress (King et al., 1994):

$$\Delta_{CFS} = \Delta\tau - \mu' \, \Delta\sigma_n \qquad (2)$$

with $\mu'$ the fault effective friction coefficient, and $\Delta\tau$ and $\Delta\sigma_n$ the shear and normal stress perturbations on the fault. The fault-slip style associated with the stress perturbation is given by the rake ($r$):



$$r = arctan\left(\frac{\tau_d}{\tau_s}\right) \qquad (3)$$

with $\tau_d$ and $\tau_s$ the shear stresses in the fault dip and azimuth directions, respectively.

In a first step, we consider the Coulomb Failure Stress perturbation to discuss the tendency of present-day GIA effects to promote slip ($\Delta_{CFS} > 0$) or inhibit slip ($\Delta_{CFS} < 0$) on a subset of fault systems in the Western Alps (a full Coulomb Failure Stress analysis integrating region stress regimes is discussed in Section 4.3). We consider three fault structures directly below the former icecap (Fig. 4a): the Vuache Fault, the Belledonne Fault, and a series of unknown faults associated with normal-faulting earthquakes in the Briançonnais directly east of the Penninic Front (hereafter IMNF, Inner Massifs Normal Faults).

The fault geometries are from neotectonic databases (Grellet et al., 1993; Jomard et al., 2017) or assuming Andersonian geometries for the unknown structures. Due to the fault dip uncertainties, we test a large range of dip angles (25–90˚). Seismicity associated with these faults concentrates in the upper crust ca. 5–10 km depth (Mathey et al., 2021; Thouvenot et al., 2003a), associated with bending stresses in the upper half of the elastic plate (cf. Section 4.1).

Because of the present-day GIA stress perturbations correspond to horizontal compressive stresses, faults with near-
vertical dip angles (80–90˚) and standard effective friction coefficient ($\mu' = 0.6$) tend to be clamped and are associated with negative $\Delta_{CFS}$ values (i.e., slip inhibition). In contrast, faults with low effective friction ($\mu' = 0.1$) or medium to low dip angles mostly show positive $\Delta_{CFS}$ between 0.1 and 2 MPa, indicating a tendency of GIA to promote fault slip in these cases. However, these models are associated with fault-slip rakes corresponding to mostly reverse faulting ($r = 50$–$130˚$, Fig. 4b), opposite to that of earthquake normal-faulting mechanisms along the IMNF ($r \approx -90˚$, Fig. 4b, Mathey et al., 2021) and at odd with the
strike-slip mechanisms on the Belledonne and Vuache Faults ($r \approx 180˚$ and $r \approx 0˚$, respectively, Fig. 4b, Rabin et al., 2018; Thouvenot et al., 2003). Thus, in all cases, the present-day GIA stress perturbations do not favor the observed seismicity on the main fault structures of the northern French Alps below the former LGM icecap.

The impacts of GIA along and outside the LGM icecap margin are more difficult to assess for two main reasons: present-day GIA stress perturbations are much smaller and, mostly, they are strongly sensitive to the model parametrization
(elastic plate thickness, rheology assumptions). As an example, we estimate the impact on the Moyenne Durance Fault, one of the best characterized active faults in the southern French Alps (Cushing et al., 2008). There, present-day GIA stress perturbations correspond to horizontal NE-SW tension ranging from ca. 0.3–0.7 MPa at the northern end of the fault to ca. 0.1–0.3 MPa at its southern end (Fig. 5a). $\Delta_{CFS}$ are systematically positive (for a low effective friction) but, due to bends in the fault segments, the range of predicted rakes is very large: along the northern section, GIA favors right-lateral strike-slip, while
favored rakes flip between right- and left-lateral along the central and southern sections (Fig. 5b). Focal mechanisms along the Moyenne Durance Fault correspond to an overall left-lateral kinematics (Cushing et al., 2008), indicating that the present-day GIA stress perturbations may favor or inhibit slip depending on the considered segment.





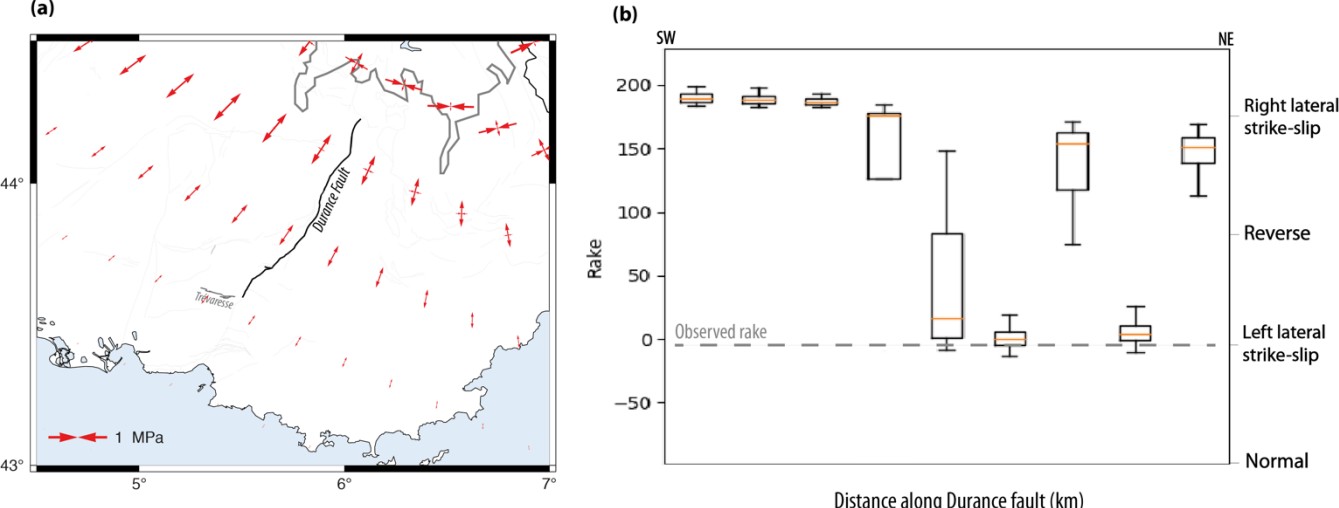

**Figure 5. Present-day GIA stress perturbations and predicted fault-slip rakes in the southern Western Alps**. (a) Horizontal GIA stress
perturbations (model $h_e$ = 20 km, $\tau$ = 5500 yr) and fault tested in the Coulomb Failure Stress perturbation analyses. (b) Fault-slip rakes
predicted by GIA stress perturbations (boxplot) and observed from earthquake focal mechanisms (Cushing et al., 2008). Boxplots represent
predictions for *72* GIA models distributed in ($5 \le h_e \le 40$ km) and ($3000 \le \tau \le 7000$ yr), combined with three possible fault dip (60°,80°,85°)

## 4.3 Present-day full Coulomb Failure Stress

The combination of GIA stress perturbations with the background stress field defines the full Coulomb Failure Stress
(*CFS*) for any given fault:

$$CFS = \tau - \mu' \, \sigma_n \tag{4}$$

with *μ'* the fault effective friction coefficient, and *τ* and *σ_n* the shear and normal stresses acting on the fault, with the associated
fault-slip rake given by eq. (3). Depending on the geometric relationships between the stress perturbation, the background
stress tensor, and the fault geometry, the *CFS* indicates that a fault should (*CFS* > 0) or should not (*CFS* < 0) slip in the local
deformation regime. A major difficulty in computing *CFS* is the definition of the background stress tensor. Given the lack of
direct measurements at seismicity depths (5–20 km), this requires several assumptions that can strongly influence the *CFS*
estimation. A second difficulty in the case of the Western Alps is the strong variations in the deformation and stress regimes
over distances of a few 10s km (Mathey et al., 2021; Delacou et al., 2004).

In order to test the conclusions drawn from the *Δ_CFS* analysis (Section 4.2), we compute the full *CFS* on two major
fault systems representative of the central Western Alps present-day seismicity: the northern Briançonnais normal-faulting
system (IMNF) and the strike-slip Belledonne Fault (Fig. 2b). In each case, we construct a local background stress tensor using
information from the focal mechanism stress inversions of (Mathey et al., 2021) and several additional assumptions for the
IMNF and Belledonne systems:

- the deformation regime is pure normal faulting (IMNF) or pure strike-slip faulting (Belledonne);





- the stress tensor is Andersonian, with the vertical stress ($\sigma_V$, lithostatic pressure) corresponding to $\sigma_1$ (IMNF) or $\sigma_2$ (Belledonne);
- the azimuth of the maximum horizontal stress $\sigma_H = \sigma_2$ (IMNF) or $\sigma_H = \sigma_1$ (Belledonne) is given by the focal mechanism stress inversion;

- the differential stress ($\sigma_1 - \sigma_3$) is controlled by optimally oriented faults with an effective coefficient of friction $\mu'$ (eq. 4) and following the Mohr-Coulomb friction law for a normal (IMNF) or strike-slip (Belledonne) case (Jaeger and Cook, 1979);
- the shape of the stress tensor ($R = (\sigma_1 - \sigma_2) / (\sigma_1 - \sigma_3)$) is set to $R = 0.5$ (i.e., $\sigma_2 = 1/2 (\sigma_1 + \sigma_3)$).

Results of the full *CFS* computation (local background stress + GIA perturbation) at 5 km depth for the northern


Briançonnais extension system are summarized in Figure 6. Assuming that the seismicity is associated with the reactivation of the Penninic Frontal Thrust (Bilau et al., 2021; Sue and Tricart, 1999), the variations of obliquity of the stress tensor to the fault geometry results in negative *CFS* from ca. -50 MPa to -10 MPa for a standard friction $\mu' = 0.6$ (Fig. 6b). A small friction $\mu' = 0.1$ results in smaller negative *CFS* (ca. -10 to 0 MPa). Relative to the modeled stress field, the northeastern

section of the fault system is more favorably oriented than southern section and shows the largest (less negative) *CFS*, up to $CFS \approx 0$ MPa in the most favorable case ($\mu' = 0.1$, fault dip of 60°). In contrast, if we assume that the seismicity is associated with faults corresponding to the average focal mechanism (azimuth N050, dip 50°; Mathey et al., 2021), this fault geometry is close to optimal orientation relative to the stress tensor and yields small, mostly negative *CFS* (ca. -5 to +1 MPa, Fig. 6c). In a few specific cases, the GIA stress perturbation pushes the fault to small positive *CFS* < 1 MPa (south-western fault section,

$\mu' = 0.1$).

In nearly all cases, the GIA stress perturbations tend to diminish the *CFS* (render more negative) by a few MPa, which confirms the tendency of present-day GIA to inhibit the normal-faulting seismicity in the Briançonnais region of the Western Alps. This is easily explained by considering that the GIA stress perturbations correspond to horizontal compressive stresses that tend to inhibit normal faulting (Section 4.2). A few specific cases can result in a positive effect of the GIA perturbation

on *CFS*, if the fault is nearly optimally oriented and the GIA stress perturbation correspond to a small subset of the tested parameter range.



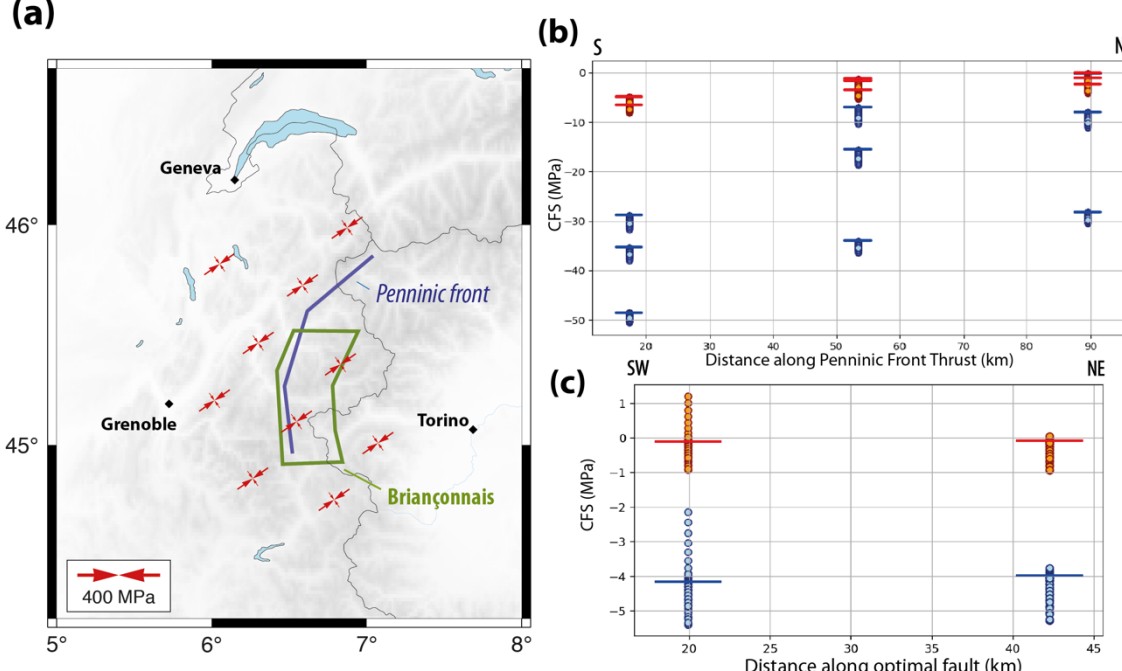

**Figure 6. Present-day full stress tensor and Coulomb Failure Stress (CFS) for the Briançonnais shallow seismicity cluster in the**
**northern Western Alps.** (a) Horizontal full stress (background + GIA) and faults tested in the CFS analyses. (b) CFS predicted along the
Penninic Front structure for three possible fault dip (30°, 45°, 60°) and 72 GIA models distributed in ($5 \leq h_e \leq 40$ km) and ($3000 \leq \tau \leq 7000$
yr). Horizontal bars: CFS without GIA perturbation for the three fault dips. (c) CFS predicted along optimal orientated fault structure in
Briançonnais area (dip = 50°) for 72 GIA models distributed in ($5 \leq h_e \leq 40$ km) and ($3000 \leq \tau \leq 7000$ yr). Horizontal bars: CFS without
GIA perturbation for the three fault dips. Circles: Full CFS (background + GIA). Blue and red symbols: $\mu' = 0.6$ and $\mu' = 0.1$.


The case of the Belledonne Fault system shows more variability (Fig. 7). The local background stress is associated
with a $\sigma_H$ azimuth of N107°, strongly oblique to the strike of the Belledonne Fault. Thus, a standard friction $\mu' = 0.6$ yields
large negative *CFS* (ca. -80 to -60 MPa at 5 km depth, Fig. 7b), whereas a small friction $\mu' = 0.1$ brings the fault - stress
configuration closer to an optimal geometry with smaller negative *CFS* (ca. -10 to -8 MPa, Fig. 7b). The impact of the GIA
stress perturbations depends on the assumed fault friction, dip and position within the local stress field: For a standard friction
$\mu' = 0.6$, the GIA perturbation systematically reduces the *CFS*, thus further inhibiting fault slip; for a small friction $\mu' = 0.1$,
the GIA perturbation can increase (render less negative) the *CFS* by 0.1–0.5 MPa if the fault dip is near vertical.

These two examples illustrate the complexity of full *CFS* estimations and their dependency on the fault and
background stress parameters. Yet, their results generally confirm the main conclusion drawn from the $\Delta_{CFS}$ analysis (Section
4.2): the present-day GIA stress perturbation tends to inhibit fault slip for the extension and strike-slip systems of the central
Western Alps (below the former icecap). A few specific configurations of background stress, fault geometry, and fault friction
can result in cases in which the GIA perturbation promotes fault slip. These particular configurations only apply to a very
limited set of fault and earthquakes.




The present analysis was carried out for a depth of 5 km, assuming that the GIA perturbation is maximum (i.e., top
of the elastic plate, Section 4.1). A smaller GIA perturbation (i.e., deeper in the upper half of the elastic plate) and a deeper
analysis depth would simply result in a smaller impact of the GIA on the full stress field and full *CFS* computations. It is also
worth noting the case of the deeper extension seismicity cluster (10–20 km depth) east of the Briançonnais cluster (cf. Section
2, Fig. 2b), for which the GIA stress perturbation would be opposite to that at shallower depths (horizontal tension, Section
4.1) and would likely favor the normal-faulting seismicity.

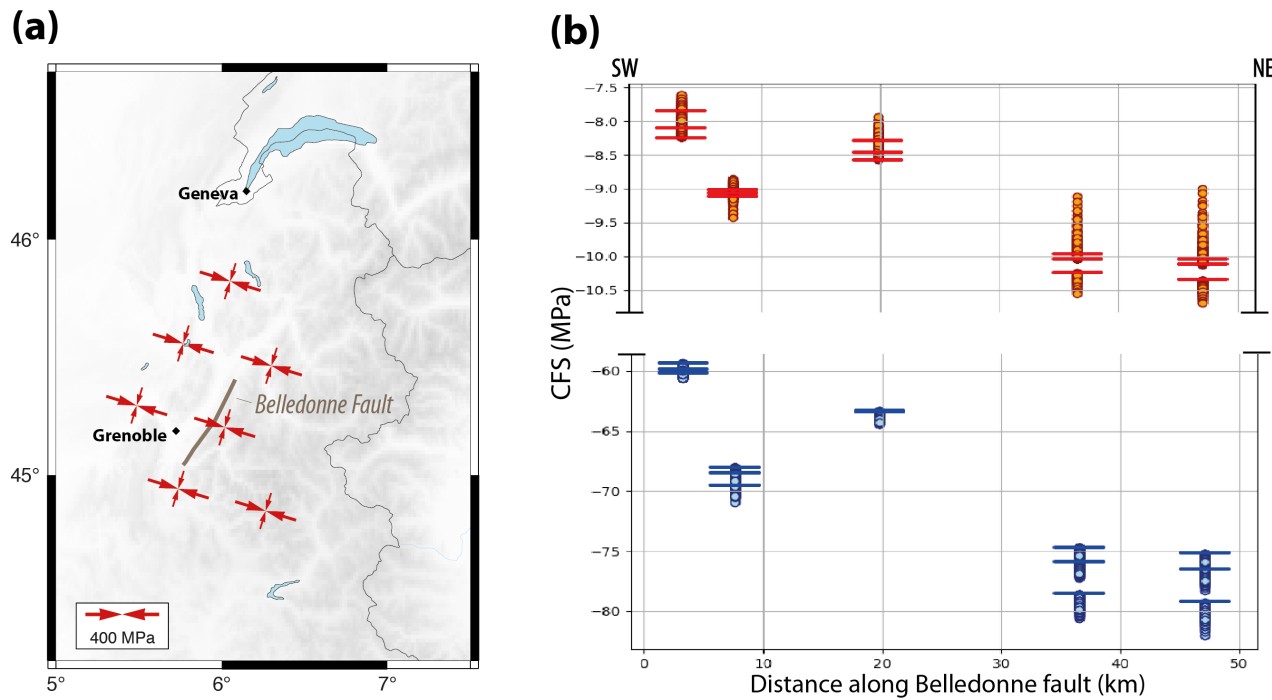

**Figure 7. Present-day full stress tensor and Coulomb Failure Stress (CFS) for the Belledonne Fault system in the northern Western
Alps.** (a) Horizontal full stress (background + GIA) and fault segments tested in the CFS analyses. (b) CFS predicted along the Belledonne
Fault for three possible fault dip (65°, 70°, 85°) and 72 GIA models distributed in (5 ≤ $h_e$ ≤ 40 km) and (3000 ≤ τ ≤ 7000 yr). Horizontal
bars: CFS without GIA perturbation for the three fault dips. Circles: Full CFS (background + GIA). Blue and red symbols: μ' = 0.6 and μ'
= 0.1.

## 5. Discussion – Why GNSS deformation rates should not be directly compared to seismicity and fault slip rates in the Western Alps

As shown by several studies in the last decade, GNSS velocities and strain rates in the Western Alps are characterized,
to a first order, by uplift rates ca. 1–2 mm yr$^{-1}$ and radial extension rates ca. (1–3) x 10$^{-9}$ yr$^{-1}$ (Masson et al., 2019; Nguyen et
al., 2016; Nocquet et al., 2016; Sánchez et al., 2018; Walpersdorf et al., 2018). These deformation rates are compatible with
the GIA effects from the Last Glacial Maximum, provided the Western Alps region behave mechanically as a thin elastic plate
over a low-viscosity upper mantle (cf. Section 3, Chéry et al., 2016; Mey et al., 2016). Although the exact contribution of GIA
relative to other processes (e.g., slab breakoff, erosion, tectonics) in the total deformation rates is unclear, the good first-order



agreement between GIA model predictions and the observed GNSS rates indicates that the former contributes to 30–80 % of
the latter, at least in the region below the former icecap (cf. Section 3, Sternai et al., 2019).

The increasing resolution of GNSS horizontal velocities and strain rates has led to direct comparisons of geodetic
styles and rates with those of earthquake focal mechanisms . In most cases, these comparisons address the existence or absence
of significant aseismic deformation rates captured by GNSS data, with direct implications on the geodynamics of the Western
Alps but also eventually on the understanding of seismicity and the characterization of the associated seismic hazard.
Ultimately, slip rates on specific seismogenic faults can be inferred using standard "plate-boundary" interseismic fault-loading
models (Mathey et al., 2020).

Yet, our study shows that, for most faults in the Western Alps, the present-day GIA stress perturbations actually tend
either to inhibit fault slip or to promote fault slip with the wrong mechanism compared to the seismicity deformation style.
Thus, at present-day in the Western Alps, GIA from the Last Glacial Maximum does not drive nor promote the observed
seismicity, which must be driven by other processes.

This apparent paradox between "GNSS-GIA strain-rate agreement" and "seismicity-GIA stress disagreement" is
easily resolved by considering that observed GNSS extension rates are merely a diminution over time of the finite shortening
induced in the upper crust by its downward flexure under the LGM icecap. The gradual diminution of the finite shortening
(i.e., extensional strain rate) corresponds to the transient return to the pre-ice situation controlled by the upper mantle relaxation
time. This situation bears similarities to that observed in forearcs of subduction zones where finite stress and deformation
styles can differ drastically from the transient strain rate patterns observed with GNSS data, the latter being due to the
interseismic locking of the subduction fault (Mazzotti et al., 2002; Wang, 2000). A similar situation exists for the early-
postglacial reverse-faulting earthquakes in Fennoscandia, which occurred during a period of GIA extension strain rates, and
hence must be the expression of compressive stress stored in the lithosphere over a long (tectonic) time (Craig et al., 2016;
Muir-Wood, 2000).

This GIA strain rate / stress antinomy has a major implication for not using GNSS velocities and strain rates in direct
comparisons with seismicity styles and rates. Because GIA does not promote the current seismicity, its surface expression,
captured in GNSS velocities and strain rates, does not have a simple relationship with seismic moments rates or fault slip rates.
In other words, GNSS velocities and strain rates comprise a significant part of transient GIA deformation that does not directly
contribute to the observed seismicity, but rather modulates the expression of the mechanisms (e.g., erosion, slab tear, tectonics)
driving the current Western Alps geodynamics and seismicity. In this situation, GNSS data cannot inform models based on a
steady-state tectonic process (e.g., far-field fault loading), but they can provide important constraints for models combining
long-term forcing and GIA transient to study fault slip and seismicity variations during and after glaciations (e.g., (Steffen et
al., 2014; Hetzel and Hampel, 2005)). This requires specific models integrating the complexities of the Alpine icecap history,
of the regional crust and mantle rheology heterogeneities, and of the local fault characteristics.



**Author contribution**

JG computed models and interpreted results. JG, SM and PV discussed the results and wrote the article.

**Competing interests**

The authors declare that they have no conflict of interest

**Acknowledgments**

This work was supported by EDF and SIGMA-2 Research and Development Project (project DEFORM_3D). Figures were made with the Generic Mapping Tool (Wessel et al., 2013) and Python 3 software. We thank T. Camelbeeck, Y. Klinger, and B. Lünd for their reviews and suggestions on early versions of this manuscript, R. Steffen for an early discussion about this work, J. Mey for providing his numerical Alpine icecap model, and M. Gamelin for her 1[st] year M.Sc. internship during
the initiation of this work

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
