# Peer review of "Glacial isostatic adjustment strain rate - stress paradox in the Western Alps, impact on active faults and seismicity"

_EGUsphere, 2023_

## Author Comment (AC1)

RC #1

In their manuscript the authors deal with the effects of glacial isostatic adjustment (GIA) on the stress and deformation patterns in the Western European Alps. In particular they investigate whether present-day observations of strain rates measured with GNSS and earthquake mechanisms correlate with the theoretical deformation pattern that would be associated with GIA. Further they investigate whether GIA would promote or inhibit movement along some of the major fault systems in the Western Alps.

They use the LGM ice load, a simplified deglaciation history and a thin-plate model with a ranges of values for the effective elastic thickness of the lithosphere (he) and upper mantle relaxation times (tau). They find that model derived strain rates are consistent with the GNSS observations in the inner Western Alps, both, in orientation and magnitude. In the foreland regions to the west and to the north only the orientation matches the GNSS observations, whereas in the south neither orientation nor magnitude are in line with the data.

Concerning the faults, they perform a Coulomb Failure Stress analysis that includes (1) only the stress perturbations caused by GIA and (2) the full stress field (GIA + background). They present results for different fault dip angles and friction coefficients and conclude that the present influence of GIA tends to inhibit fault slip and that the observed earthquake kinematics is at odds with the deformation predicted for GIA and measured with GNNS.

Their main conclusion is that the GNSS is dominated by transients caused by GIA, whereas the seismicity reflects long-term geological forcings.

The manuscript is well written and I have no objections with it being published except that it lacks a conclusion section and the figures should be improved.

A conclusion section has been added at the end of the article.

**Minor comments/edits:**

*Line 103: flowing -> following*

Corrected l.111.

*Line 186: high altitude*

Reformulated using "northern", l. 200.

*Line 191 – 200: Is there an effect on the strain rate induced just by the topographic gradient between the Alps and the foreland?*

This effect hasn't been tested in the frame of this study. Although this hypothesis is of interest, studies on European mountains suggest a negligible effect of gravitational collapse

on geodetic fields and present-day deformation in the Alps (Hivert et al., 2011; Vernant et al., 2013).

*Line 222: in mind*

Corrected l.240.

*Line 234: IMNF should be defined at its first use*

Corrected, definition added l.254.

*Line 240: I suggest to use tau either for the relaxation time or the shear stress, not for both. Maybe add a subscript.*

Corrected. Relaxation time has been changed to "$\tau_r$" in the text and figures.

*Line 254: delete "of"*

Corrected l. 275.

*Line 283: µ' has been defined already in Line 241*

Corrected l. 308.

*Line 327: oriented*

 Corrected l. 364.

Figures:

The figures could be improved by using only one font style and size and a more consistent panel labeling.

- *add scale bars to all the maps*

Scale bars has been added on figure 2, 3, 4, 5, 6 and 7.

- *Fig. 3. omit the dot in the velocity unit*

As the standard formulation "mm.yr$^{-1}$" is used throughout the text, we prefer to keep the dot in the velocity unit in figure as well.

- *Fig. 6 & 7: I struggled with the symbology of panels b) and c). The results for the different dip angles cannot be distinguished. Perhaps different marker symbols could be used (e.g. squares, triangles etc.). I suggest to replace the horizontal bars with markers.*

We agree that it would be interesting for the details of the parametrizations to appear in the figures. Unfortunately, adding this information makes the figure unreadable. Thus, details of the parameters corresponding to an increase in the Coulomb stress on an optimally oriented fault (fig. 6.c) have been added to the text (l. 348).

Furthermore, our study emphasizes the general impact of the GIA in the Western Alps. Thus, a more detailed parametric study on the projection of stress perturbations on specific faults should be realized thereafter.

- *Fig. 6: "(a) Horizontal full stress (background + GIA) and **faults** tested in the CFS analyses." But there is only a single fault shown in (a).*

Corrected l. 362.

Hivert, F., Vernant, P., chery, J., Cattin, R., and Rigo, A.: Can the gravitational collapse paradigm withstand the geodetic and seismologic observations in the Alps and the Pyrenees, birth of a new paradigm?, 2011, EP41D-0636, 2011.

Vernant, P., Hivert, F., Chéry, J., Steer, P., Cattin, R., and Rigo, A.: Erosion-induced isostatic rebound triggers extension in low convergent mountain ranges, Geology, 41, 467–470, https://doi.org/10.1130/G33942.1, 2013.

---

## Author Comment (AC2)

**RC2**

Dear Editor and authors,

Please find enclosed my review of the manuscript "Glacial isostatic adjustment strain rate - stress paradox in the Western Alps, impact on active faults and seismicity" by Grosset, Mazzotti & Vernant. The manuscript investigates the effects of the latest Alpine glaciation on current seismicity in the Western Alps, using models of glacial isostatic adjustment (GIA), current earthquake mechanisms and the current stress field as inferred from the earthquake mechanisms. The manuscript is a revision of an earlier version, which I find is significantly improved and ready for publication with some minor revision.

Comments and suggestions:

Abstract

*- The authors use the term Last Glacial Maximum (LGM) as a name for the latest major European glaciation, both in the Abstract and elsewhere in the manuscript. The LGM is a specific point in time, not a name. The latest Fennoscandian ice sheet is known as the Weichselian glaciation, I am not sure if this includes the Alpine ice cap, or if that has another name, but the authors should not use LGM in a naming sense.*

Every occurrence of LGM referring to an ice cap have been removed and replace by the ice cap name (e.g. Würm for the alpine ice cap)

*- I am not sure that the first sentence: "In regions formerly glaciated during the Last Glacial Maximum (LGM), Glacial Isostatic Adjustment (GIA) explains most of the measured uplift and deformation rates." is correct. Does this really apply to active regions like Alaska or Western Canada?*

Agreed, this sentence cannot be strictly applied to Alaska and Western Canada. We corrected the abstract by adding « many ».

Line:

*- 25. Deglaciation increases seismicity in regions with appropriate tectonic background stress, such as cratons or regions in a reverse stress state. It is likely to decrease seismicity in regions with normal stress, such as e.g. in Iceland, see e.g. Lund (2015). I suggest modifying the sentence with a "... in many regions formerly..."*

Corrected l.25. The reference Lund (2015) has been added l. 27.

*- 29. Grollimund & Zoback, 2001 unfortunately has methodological problems, I would not refer to that.*

This reference has been removed from l. 29.

*- 44-45. Naming of the glaciation don't use LGM.*
Corrected l. 44 and 46.

*- 70. The sentence is a bit confusing, the events can be divided into two N-S clusters, one in the west and one in the east? The cluster seem elongated in N-S, just clarify the writing a little.*

This sentence has been clarified l. 71.

*- 103. Spelling "flowing" -> "following"*

Corrected l. 104.

*- Section 3 on GIA models and Figure 3. I would like to see at least a map of the modelled uplift rates, at the same locations as the GNSS data points in Fig 3a, or a map of the difference in these points. As the uplift rates are used to evaluate model fit, this would help the reader assessing how good the models are.*

A map of GNSS and modeled vertical velocities has been added in supplementary material (fig. S5) and the reference of this figure l. 171.

*- 155. In the GIA modelling the authors assume isostatic equilibrium at the LGM owing to the short relaxation time of the mantle. But then they use relaxation times between 2,000 and 20,000 years in the modelling. A Maxwell relaxation time of 2,000 years corresponds to a viscosity of about $6x10^{(21)}$ Pa s (for Young's modulus $10^{(11)}$ Pa), which is a fairly high viscosity mantle. 20,000 years corresponds to lower mantle viscosities. There is therefore a discrepancy between the used relaxation times and the assumption of isostatic equilibrium at LGM.*

The formulation wasn't clear in the text. We reformulate by precising the best-fit relaxation time (ca. 5000 yr) l. 156, which justifies the assumption of isostatic equilibrium. The larger range of tested relaxation time is detailed in section 3.2 (GIA velocities and strain rates), l. 170-175.

*Also, I would have thought that the Alpine mantle would have lower viscosity/relaxation time. In the Supplement the authors model an ice sheet which resides on the model for 100 kyr before instantaneously melting. It is unlikely that the LGM lasted for such a long time, even the Fennoscandian ice sheet only had (almost) LGM dimensions for a few thousand years, which is not enough to reach equilibrium. The authors should at least discuss this.*

Indeed, the alpine ice cap variated during last 100 ka (Ivy-Ochs et al., 2008). However, the goal of this test is to compare the "1D Maxwell" and thin elastic plate models, in order to show the validity of our approach. For this purpose, we test a simple case considering a lithosphere that reach equilibrium after the glaciation.

Ice cap time variations may have a small present-day impact depending on the detailed glaciation history and Earth rheology. These should be considered in further study.

*- 165. It seems the model testing in the supplement is performed on a relatively small (horisontally) model in comparison to the size of the ice sheet. This is likely to affect the stress values.*

The horizontal axes on schematic representation of 1D Maxwell model set up (Fig. S2) has been forgotten. We corrected the x axis in figure S2: the model is actually 4000 km wide, large enough to avoid border effects with the (relatively small) 200-km-wide ice cap.

*It is also unclear if the GIA/viscoelastic model takes into account stress advection.*

Yes, stress advection is built in the "1D Maxwell" model, through the assumed visco-elastic Maxwell rheology and the explicit integration of gravity (cf. visco-elastic gravitional relaxation in Wu (1992)).

*In addition, it would have been advantageous with plots of the stress at the bottom of the elastic plates, in order to compare how stress is modeled there (for the discussion in line 348).*

We added a figure (Fig. S3.c) to show the bending stress at the base of plate. As hinted by the reviewer, this comfort our thin elastic plate approximation.

*- 241. Perhaps add that you follow Bott (1959) in assuming slip in the direction of maximum resolved shear stress on the fault.*

This reference has been added in the text, l. 244.

*- 300 and below. Indicate how you use the coefficient of friction. Do you always have the same mu when constructing the stress state as when evaluating CFS? I think that is a good first order assumption, but one could also think that established faults have a lower mu and slip more often.*

The same coefficient of friction has been used for stress state construction and stress perturbation computation in this study (added in the text l. 307).

It should be interesting to study different friction between faults and background stress, that we will done in further study.

*- 321. I find it interesting that you find that a normal stress background field can in some cases be pushed into failure by a reverse GIA induced field, under the former ice cap. Could you deliberate bit more on this; angle between fault and SHmax, stress magnitudes, dip and rake in these cases?*

GIA models that predict an increase in Coulomb stress are characterized by really small elastic thickness (≤10km) because of the obliquity of GIA stress SHmax (azimuth = N90˚) compared to the fault azimuth (N050˚), involving an increase in tangential stress in the fault. The range of elastic thickness triggering that case has been added l. 319.

*- 375. Add to the discussion, here somewhere, that the models predict that in the future, with diminishing GIA stress components, seismicity will increase as the crust comes back to more normal/strike-slip stress conditions.*

We added a sentence to precise this point l. 345-347.

*- 403. Thanks for having me in the Acknowledgment! I don't have the umlaut on my surname, it is just "Lund".*

Corrected l.424.

Figures:

*- Figs 4, 5, 6, 7. I prefer to illustrate the stress field with lines/bars of the same size showing the direction of SHmax, and colour contours of the magnitude of SHmax, instead of the double arrows used in (a) in these figures. That more clearly shows how SHmax rotates 90 degrees outside of the former ice margin, and better indicates the magnitude. Especially in Fig 4, the double arrows outside the ice margins are so small that they are virtually impossible to interpret.*

We agreed that the orientation of small stress vectors is difficult to read. Nevertheless, we prefer to represent stress field that way to keep visible the difference between strike slip and uniaxial horizontal stress. In order to clarify the figure, the stress scale has been increased in figure 4.

*- Figs 5, 6, 7. It took me a while to find the indicators of "SW", "NE" and similar on top of the CFS plots. I would remove those and write in the xlabel "Distance from SW to NE along..."*

Xlabel on figure 5, 6 and 7 has been corrected.

*- Add explanation of the orange line in the box plots.*

The meaning of the orange line has been added to Fig. 4 and 5 legends.

Reference: Lund, B. (2015) Palaeoseismology of glaciated terrain. In Beer, M., Kougioumtzoglou, I.A., Patelli, E., Au, S-.K. (eds.), Encyclopedia of Earthquake Engineering, Springer Berlin Heidelberg, 1765-1779, doi: 10.1007/978-3-642-36197-5_25-1.

Ivy-Ochs, S., Kerschner, H., Reuther, A., Preusser, F., Heine, K., Maisch, M., Kubik, P. W., and Schlüchter, C.: Chronology of the last glacial cycle in the European Alps, Journal of Quaternary Science, 23, 559–573, https://doi.org/10.1002/jqs.1202, 2008.

Wu, P.: Viscoelastic versus viscous deformation and the advection of pre-stress, Geophysical Journal International, 108, https://doi.org/10.1111/j.1365-246X.1992.tb00844.x, 1992.